# Tackling Sparse Facts for Temporal Knowledge Graph Completion

## ABSTRACT

Temporal knowledge graph completion (TKGC) seeks to develop more comprehensive knowledge representations by addressing missing relationships and entities within temporal knowledge graphs (TKGs), thereby enhancing reasoning and predictive capabilities in downstream tasks. Nonetheless, real-world knowledge—such as the progression of social network interactions and the unfolding of news events—is inherently dynamic, resulting in substantial sparsity issues in TKGs that profoundly impair the performance of TKGC models. To overcome this challenge, we introduce the **A**daptive **N**eighborhood **E**nhancement **L**ayer (ANEL), a novel module that can be effortlessly integrated into existing TKGC models to substantially elevate the representation quality of sparse entities. ANEL first derives initial entity embeddings through a base model and then uncovers concealed semantic relationships between entities via a latent relation module, enriching the explicit relationships within the knowledge graph. Furthermore, ANEL incorporates an adaptive latent information adjustment component, which dynamically calibrates the influence of latent information based on the entity's relational structure: entities with fewer connections derive greater benefit from latent information, while entities with denser connections become less dependent on latent augmentation, ensuring precise and resilient representations. We conducted comprehensive experiments on four prominent benchmark datasets, and the results underscore the effectiveness and superiority of ANEL in TKGC tasks. The code is available at: https://anonymous.4open.science/r/ANEL-177F.

## KEYWORDS

Knowledge Representation, Temporal Knowledge Graph Completion, Fact Sparsity

**ACM Reference Format:**
Anonymous Author(s). 2024. Tackling Sparse Facts for Temporal Knowledge Graph Completion. In *Proceedings of The Web Conference 2025 (WWW '25)*. ACM, New York, NY, USA, 10 pages. https://doi.org/XX.XXX/XXX

## 1 INTRODUCTION

Knowledge graphs (KGs) utilize a graph-based framework to reveal intricate relationships within data, emerging as vital tools for the structured representation and organization of knowledge. By encoding real-world information in the form of triples—comprising a subject entity, a relation, and an object entity—KGs construct a network that facilitates the in-depth analysis of complex inter-entity

interactions. This architecture endows KGs with formidable semantic representation and reasoning capabilities, significantly boosting performance across various downstream tasks. Their applications span a wide array of fields, including question-answering systems [1], recommendation engines [2–4], and information retrieval systems [5]. For instance, in question-answering systems, KGs enable more precise and contextually informed responses through enhanced reasoning; in information retrieval, they improve the accuracy and relevance of search outputs; in recommendation engines, they analyze user behavior and preferences to generate more personalized suggestions. However, in practice, KGs often grapple with significant information gaps and incomplete data, which can severely hamper their ability to faithfully represent real-world knowledge, thereby diminishing their efficacy in various applications. To address this challenge, KG Completion (KGC) plays a crucial role [6]. KGC employs machine learning and inference techniques to predict and fill in missing triples, thereby enriching and expanding KGs while bolstering their performance across a wide range of applications.

KGC, as a critical downstream task within KGs, has garnered significant attention from the research community due to its focus on inferring missing facts by identifying patterns and rules within existing KG data. A variety of KGC approaches leverage KG embeddings (KGEs) to project high-dimensional data into lower-dimensional spaces, facilitating the creation of a mapping function between entities and their relationships. This mapping function is then used to evaluate the plausibility of predicted facts. While these methods demonstrate strong performance in predicting missing facts, real-world knowledge is often subject to temporal shifts, such as those observed in news updates and evolving social dynamics. To maintain accuracy, each dataset is typically annotated with timestamps. For instance, Donald Trump served as President of the United States from November 9, 2016, to January 11, 2021 *(Trump, Is President of, United States, 20161109-20210111)*. Addressing these temporal changes, Temporal KGs (TKGs) have been introduced, linking events to specific timestamps. However, the recurrence of similar events in different temporal contexts creates distinct facts. For example, the ICEWS dataset [7] records the same event with different timestamps: *(Angola, Make a visit, China, 2014-06-06)* and *(Angola, Make a visit, China, 2015-06-23)*. This added temporal complexity poses substantial challenges for the task of temporal KGC (TKGC).

Despite the significant advancements made by current TKGC models, accurately predicting entities with sparse relationships or newly introduced entities within TKGs remains a formidable challenge. We conducted an in-depth analysis of widely utilized benchmark datasets in the TKG domain, such as ICEWS14 [8], ICEWS18 [9], GDELT [10], and ICEWS05-15 [9], with particular attention to the impact of neighboring entities on prediction performance (e.g., MRR). Using the REGCN [11] TKGC model, we evaluated entity prediction across various temporal subgraphs under different neighbor counts. To consolidate our findings, we categorized entities by their

neighbor counts and calculated the average MRR for each group, leading to the final analytical insights. As shown in Figure 1, the blue bar chart illustrates the distribution of entities by neighbor count within TKGs, revealing that entities with only one neighbor form the largest group, emphasizing the pervasive sparsity within TKGs—a limitation that restricts the available information for fact completion. The accompanying light blue bar chart depicts the relationship between neighbor count and the prediction performance of entities with missing facts. Our analysis demonstrates a positive correlation between the number of neighbors and enhanced prediction accuracy, underscoring the crucial influence of neighbor count on prediction outcomes.

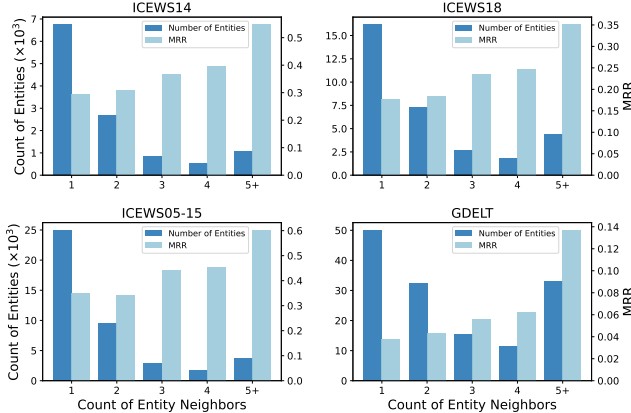

Figure 1: The influence of varying entity neighbor quantities on entity prediction efficacy across the ICEWS14, ICEWS18, ICEWS05-15, and GDELT benchmark datasets utilizing the REGCN model.

In this work, we propose a model-agnostic **A**daptive **N**eighborhood **E**nhancement **L**ayer (ANEL) specifically designed to address the challenge of fact sparsity in TKGs. This enhancement layer can be seamlessly integrated with any TKGC model, improving the representation of entities with limited neighbors and consequently boosting prediction accuracy for missing facts. When entities have a large number of neighbors, the framework enhances their representations by aggregating information from surrounding entities; in contrast, when neighbors are sparse, the enhancement layer leverages latent neighbor information to enrich these entities with essential data. Extensive experiments conducted on widely recognized benchmark datasets, such as ICEWS and GDELT, demonstrate that our proposed ANEL framework markedly improves entity prediction performance, especially in scenarios characterized by sparse neighbor connections. Our principal contributions are as follows:

- We systematically analyzed and quantified the influence of entity neighbor count on prediction performance within TKGs, uncovering the critical impact neighbor count has on overall model effectiveness.
- We introduced a novel, model-agnostic ANEL module that can be effortlessly incorporated into any existing TKGC model, substantially improving the representation of entities with limited neighbors.

- We carried out extensive experimental validation on four widely recognized TKG benchmark datasets, showcasing the efficacy of our proposed enhancement layer.

## 2 RELATED WORK

Existing TKGC models can be broadly divided into two categories: non-neural models and Neural network-based TKGE models [12].

Non-neural models typically extend traditional KG embedding approaches by incorporating temporal information into the representations of entities and relations, using transformed distances to assess the plausibility of facts. For instance, the t-transE [13] captures temporal information indirectly by ordering relations based on the time of occurrence; the HyTE [14] employs time information as a hyperplane to map entity-relation pairs; while the ChronoR [15] maps entities and relations into complex space, incorporating time information and linear operators for reasoning, and using rotations to infer complex relationships. Another subset of non-neural network methods uses tensor decomposition to evaluate the plausibility of quadruples, as seen in models like T-DistMult [16], TNTComplEX [17], TeLM [18], Cont [19], and TuckERTNT [16]. To address uncertainties in the representations of entities and relations and enhance the model's expressiveness, the AtiSE [20] models entity embeddings as Gaussian distributions, and the RE-NET framework [9] uses conditional probability distributions to describe the patterns of fact occurrence. Additionally, the DBKGE [21] applies the mean and variance of Gaussian distributions to entity embeddings to capture embedding uncertainties and improve representation quality. Despite being intuitively designed and relatively simple, these models often produce embeddings with limited quality due to their simplicity and limited parameterization.

Neural network-based TKGE models excel in representation learning and the capture of temporal information, effectively handling time dependencies and dynamic evolution within KGs. A key advantage of these models is their ability to reveal complex entity relationships and their evolution over time through various architectures. For instance, TKGE models combining Graph Neural Networks (GNNs) with Recurrent Neural Networks (RNNs) are among the most common architectural types. GNNs excel at capturing structural dependencies between entity nodes, while RNNs track the changes in entities and relationships over different time points. Specifically, the DACHA [22] employs a dual graph convolutional network (GCN) based on historical relationships to capture the influence of relational interactions within a graph's neighborhood and uses a self-attention encoder to model dependencies between different event types. Similarly, the TeMP [23] combines a Relational GCN (RGCN) to capture the influence of neighboring entities and employs a frequency-based gated GRU to handle dependencies among inactive facts. The REGCN [11] also utilizes RGCN to capture the effects of neighboring entity interactions, while an autoregressive GRU is used to model the associations between temporal facts. Moreover, emerging technologies such as Capsule Networks [24–26], Transformers [27, 28], Bert [29, 30], Meta-Learning [31–33], Reinforcement Learning [34, 35], and Large Language Models [36, 37] have also been widely applied to TKGC tasks, providing robust support for further enhancing model performance. Although graph neural network-based TKGC

methods can predict missing information using the neighborhood information of target nodes, sparse neighborhood information can hinder the update of entity representations and affect model performance. Zhang et al. [38] introduced a latent relation learning approach; however, it inadequately delves into the richness of existing entity information, making it susceptible to incorporating superfluous noise. Moreover, it lacks adaptability, thereby restricting its applicability across diverse models. Conversely, Mirtaheri et al. [39] proposed a TKGC model related to entity degree that affords a degree of flexibility. Nevertheless, this method overlooks the structural intricacies and contextual semantics of latent entity relations, both of which are pivotal for refining entity embeddings and bolstering predictive accuracy.

## 3 PRELIMINARY AND DEFINITION

In this section, we systematically introduce the basic definitions of temporal knowledge graphs (TKGs) and provide a detailed description of the key symbols and the process involved in TKG completion (TKGC) tasks.

TKGC involves predicting or filling in previously missing factual data as the KG evolves over time. These tasks are commonly known as extrapolation and interpolation. This paper primarily focuses on the extrapolation task within TKGs. A TKG consists of multiple graph snapshots taken at different time points $\mathbf{G} = (\mathcal{G}_1, \mathcal{G}_2, \ldots, \mathcal{G}_t)$, where each snapshot $\mathcal{G}_t = (\mathcal{F}_t, \mathcal{E}_t, \mathcal{R}_t)$ reflects the set of facts $\mathcal{F}_t$ formed by interactions between entities in the entity set $\mathcal{E}_t$ and relationships in the relationship set $\mathcal{R}_t$ at time step $t$. Each fact can be represented as a quadruple $\mathcal{F}_t = (s, r, o, t)$, where $s$ and $o$ are the subject and object entities, respectively, $r$ denotes the relationship between them, and $t$ is the timestamp of the interaction. Unlike static knowledge graphs (KGs), TKG extrapolation places significant emphasis on real-time data updates.

The TKGC model treats the prediction task as a sequence of tasks across multiple time steps $T = (T_1, T_2, \ldots, T_n)$. Each sub-task $T_t$ includes mutually exclusive training data $D_{\text{train}t} \in G_t$, testing data $D_{\text{test}t} \in G_t$, and validation data $D_{\text{val}t} \in G_t$ to ensure unbiased evaluation. The model starts by learning from the training data $D_{\text{train}1}$ at the first time step, using the validation data $D_{\text{val}1}$ to tune hyperparameters. In each subsequent time step, the model builds on the parameters from previous steps, iteratively updating them to enhance prediction accuracy for future data.

## 4 ADAPTIVE NEIGHBORHOOD ENHANCEMENT LAYER

TKGs are characterized by dynamic structures, where some entities may frequently engage in relationships during certain periods while remaining inactive during others. This temporal imbalance leads to sparse connections for certain entities at specific time steps. Additionally, as TKGs evolve, entities and relationships are continuously expanded and updated. Newly introduced entities may receive attention and connections in certain parts of the graph, while being neglected in others, resulting in insufficient connections and representations. These sparsely connected entities typically have lower-quality representations within the KGs, resulting in suboptimal performance of TKGC models when tasked with predictions involving these entities.

We propose an **A**daptive **N**eighborhood **E**nhancement **L**ayer (ANEL), a module that can be seamlessly integrated into any temporal knowledge graph completion (TKGC) model. Its purpose is to enhance the embedding representations of sparsely connected and low-activity entities by enriching their latent structural information and semantic context. First, the TKGC model is used to extract the basic features of the entities, generating their initial representations. Then, the ANEL module systematically explores and adaptively supplements the latent structural and semantic information based on the sparsity of the entity relationships. Through this process, ANEL effectively improves the base model's ability to represent sparsely connected and low-activity entities. According to the format outlined in [39], the entity representations processed by ANEL are as follows:

$$e_{s,t} = f(s) + \phi(s)g(s) \quad (1)$$

In this formulation, $e_{s,t}$ signifies the ultimate representation of the query entity $s$ at time $t$, whereas $f(s)$ denotes the embedding of entity $s$ produced by the underlying TKGC model. The entity activity function $\phi(s)$ is defined as $\frac{1}{1+\exp(|\mathcal{N}_s^t|)}$, where $|\mathcal{N}_s^t|$ represents the number of interacting neighbors of entity $s$ at time $t$, reflecting the entity's activity level at that particular moment. A lower value of $\phi(s)$ indicates heightened activity of the entity at timestamp $t$, characterized by frequent interactions with other entities. For such highly active entities, supplementary information is generally redundant due to the richness of interaction data. Conversely, for less active entities with sparse interaction data, it may be necessary to explore potential interactions to enhance the quality of their representations. The enhancement of entity representations through latent neighborhoods involves three principal stages: the construction of a latent graph $g(s)$, temporal neighbor sampling, and the aggregation of latent neighborhoods.

### 4.1 Latent graph construction

For the prediction task $(T_1 = (s, r, ?, t_1))$, we begin by generating a candidate set of potentially related entities $C_{t_1}^s = (o_i \in G_{t_1})$ by filtering out those that do not directly interact with entity $s$ at the current timestamp $t_1$, as entities appearing within the same timeframe are seldom coincidental and often exhibit latent connections. For example, events such as $(Russia, military intervention, Ukraine, 202202)$ and $(UK, aid, Ukraine, 202202)$ are likely intertwined due to their temporal alignment.

Subsequently, a base model encoder is utilized to initialize the candidate entities' features, embedding structural information into their respective feature representations, as demonstrated by the equation $h_{o_i} = \text{Encoder}(o_i)$. This step enhances the model's capacity to uncover intricate relationship patterns within the temporal graph. Finally, the Pearson correlation coefficient is utilized to quantify the linear relationships between the features of the candidate entities and the target entity, providing a more sophisticated understanding of their interconnections.

$$d(s, o_i) = \text{PearsonSimilarity}(h_s, h_{o_i}) \quad (2)$$

To refine the precision of identifying entities with potential interaction relationships and optimize the model's computational efficiency, we employ the K-Nearest Neighbors algorithm [40] to

**Figure 2: Illustration of the proposed ANEL integrated with the base model, which first generates initial embeddings, refines entity features through a latent neighbor graph, and combines the adaptive information component's output with the base model's entity features for entity prediction.**

select the top $k$ entities most similar to the query entity based on their feature embeddings. These selected entities are then integrated as new neighbors to construct the potential graph structure. The edges in the potential graph $\mathcal{G}^l$ are defined as:

$$\mathcal{G}^l = \begin{cases} 1, & \text{if } d(s,o) \in \text{top-}K(C_{t_1}^s) \\ 0, & \text{otherwise} \end{cases} \quad (3)$$

When an entity has few links at time $t$ or is a new entity, and the number of potential neighbors in the current temporal subgraph is less than $k$, we supplement the potential neighbors by searching from the previous temporal subgraph at $t-1$. Since the appearance of entities at the same time is often non-random, suggesting some underlying interaction or shared context, we prioritize selecting entities that co-occur with the target entity at the same time. This ensures that the potential neighbors are temporally relevant and likely to share meaningful interactions.

## 4.2 Latent Neighbourhood Repretation Aggregation

To fully exploit the rich relational information embedded in the potential graph, we execute relation-aware propagation and aggregation operations within the latent neighborhood graph of the target entity [38, 41, 42]. By utilizing relation-aware propagation, the target entity assimilates varied relational information from its surrounding entities, enriching its representation with enhanced contextual semantics. The entity representation, updated through the incorporation of latent neighborhood information, is expressed as:

$$\mathbf{h}_o^{(l)} = \sigma\left(\sum_{(s,r,o)\in\mathcal{G}^{(l)}} \alpha_{sro}\mathbf{W}_r^{(l)}\left(\mathbf{h}_s^{(l)} + \mathbf{h_r}\right) + \mathbf{W}_o^{(l)}\mathbf{h_o}^{(l-1)}\right) \quad (4)$$

Here, $\mathbf{h}_o^{(l)}$ denotes the embedding of entity $o$ at layer $l$, and $\mathbf{h}_o$ refers to its representation. The function $\sigma$ corresponds to the *ReLU* activation function, while $\mathbf{W}_o^{(l)}$ and $\mathbf{W}_r^{(l)}$ represent the learnable parameters for entities and relations, respectively, in the $l$-th layer of the learning process. The attention weight $\alpha_{sro}$ is used to distinguish the significance of entity $o$ in relation to entity $s$ within the potential graph. Following the approach outlined in [41], the attention score $\alpha_{sro}$ for each fact is normalized, with the softmax function applied to calculate the relative importance of entity $o$ to entity $s$ within the potential graph $\mathcal{G}^l$:

$$\alpha_{sro} = softmax(w_{sro})$$

$$w_{sro} = \frac{\exp(w_{sro})}{\sum_{w\in\mathcal{N}_s}\exp(w_{wro})}$$

$$w_{sro} = \boldsymbol{\alpha}^{(l)}LeakyReLU\left(\mathbf{W}_{att}^{(l)}\left[\mathbf{h}_s^{(l-1)}\|\mathbf{h}_r^{(l-1)}\|\mathbf{h}_o^{(l-1)}\right]\right), \quad (5)$$

$$\forall(s,o)\in\mathcal{G}^l$$

where $\boldsymbol{\alpha}^{(l)}$ and $\mathbf{W}_{att}^{(l)}$ represent the learnable parameters for the subject entity, relation, and object entity at layer $l$, respectively. The $\|$ denotes the concatenation of different vectors. The vectors $\mathbf{h}_s^{(l-1)}$, $\mathbf{h}_o^{(l-1)}$, and $\mathbf{h}_r^{(l-1)}$ represent the hidden representations of the subject entity, object entity, and relation at layer $l-1$, respectively. The set $\mathcal{N}_s$ represents the direct neighbors of entity $s$ in the graph $\mathcal{G}^l$, while $\mathcal{N}_o$ denotes the set of entities that have interacted with entity $s$ in the same graph.

## 5 EXPERIMENTS

In this section, we assess the efficacy of the proposed adaptive neighborhood enhancement layer (ANEL) for entity prediction in TKGC, utilizing four prominent TKG datasets. The research questions are as follows:

- **RQ1:** How does the performance of the TKGC model integrated with ANEL compare to baseline models?
- **RQ2:** How do variations in model weight parameters influence the performance of the TKGC model when ANEL is applied?
- **RQ3:** What are the individual contributions of the ANEL components to the overall performance in TKGC tasks?

## 5.1 Experiment Settings

*5.1.1 **Datasets.*** To validate the efficacy of our proposed ANEL, we employed four widely recognized benchmark datasets in the domain of TKGs: ICEWS14 [8], ICEWS05-15 [9], ICEWS18 [9], and GDELT [10]. The first three datasets, sourced from the Integrated Crisis Early Warning System (ICEWS), chronicle interactions between geopolitical entities for the years 2014, 2005-2015, and 2018, respectively. Conversely, the GDELT dataset records global news events from 1979 onwards, encompassing data from worldwide news broadcasts and online media. We processed the datasets following the methodology outlined in [9], partitioning each into training, validation, and test sets with an 80%, 10%, and 10% split, respectively. Detailed dataset statistics are presented in Table 1.

**Table 1: Common TKGE benchmarks and their attributes**

| Datasets | Entities | Relations | Facts | Train Facts | Time Steps |
|---|---|---|---|---|---|
| ICEWS14 | 7128 | 230 | 90730 | 74845 | 365 |
| ICEWS05-15 | 10488 | 251 | 479329 | 368868 | 4017 |
| ICEWS18 | 23033 | 256 | 468558 | 373018 | 304 |
| GDELT | 7691 | 240 | 3419607 | 1734399 | 366 |

*5.1.2 **Evaluation Metrics.*** In our experimental validation, we use the state-of-the-art TKGC model TiRGN [43], which performs best on extrapolation tasks, and the second-best performing model REGCN [11] as our base models to validate the effectiveness of our proposed ANEL method. For the evaluation metrics, we use two widely recognized measures to assess model performance in TKGC tasks: Hit@k and Mean Reciprocal Rank (MRR). Hit@k measures the percentage of cases where the correct entity appears in the top $k$ predicted results, providing insight into the accuracy of the model's top-ranked predictions. MRR calculates the average of the reciprocal ranks of correct predictions, offering a more granular evaluation of model ranking performance across all tasks. In this study, we employ Hit@1,3,10 to evaluate model performance. We report both our results and those from the baseline experiments, using the same time-aware filtering method as applied by Li et al. [43], to ensure a fair and consistent comparison.

*5.1.3 **Baselines.*** In our baseline model setup, we select several high-performing static and TKGC models as benchmarks. The static KGC models include DistMult [44], ConvE [45], RotatE [46], and ComplEx [47]. These models handle only entities and relations without considering temporal information. In contrast, the TKGC models include TNTComplEx [17], RE-NET [48], CyGNet [49], xERTE [50], TITer [51], REGCN [11], TiRGN [43] and CENET [52]. These models incorporate temporal information, allowing them to better capture and predict evolving relationships and entities over time.

- **DistMult** [44] represents relationships between entities in a KG by performing bilinear dot products between the embeddings of the head entity, relation, and tail entity.
- **ConvE** [45] captures complex relationships in KGs by transforming entity and relation embeddings into 2D matrices and applying convolutional neural networks to them.
- **RotatE** [46] models relationships by rotating the head entity vector to the tail entity vector in the complex space, representing triplet relationships in KGs.
- **ComplEx** [47] embeds entities and relations in the complex space and uses the complex dot product to model symmetric, antisymmetric, and complex relational patterns.
- **TNTComplEx** [17] is a TKGE method that represents temporal data as fourth-order tensors in the complex space, leveraging complex numbers to efficiently model the interaction between entities, relations, and time.
- **RE-NET** [48] captures dynamic interactions between entities and relations over time by using a recurrent neural network (RNN) combined with a relational graph convolutional network (RGCN) to model temporal evolution.
- **CyGnet** [49] predicts new facts by leveraging historical and cyclic events within a temporal context, focusing on the recurrence of specific event types over time.
- **xERTE** [50] is an interpretable model that predicts new events by using subgraphs and attention mechanisms to focus on relevant historical information for explaining predictions.
- **TITer** [51] is a reinforcement learning-based model that predicts future facts by selecting optimal time paths—sequences of events that unfold over time—using reinforcement learning. The model learns to navigate historical temporal data and identifies key event sequences that are most likely to influence future predictions, optimizing its path selection through iterative feedback.
- **REGCN** [11] uses a graph convolutional network (GCN) to learn structural features in KGs and combines it with an RNN to model temporal sequences, capturing dynamic relationships between entities over time.
- **TiRGN** [43] models evolving entity relationships in dynamic KGs by using a local encoder to capture dependencies between historical facts and a global encoder to capture repeating historical patterns over time.
- **CENET** [52] is a contrastive learning-based TKGC model that uses contrastive learning to decide whether historical information should be used to predict new facts.

*5.1.4 **Implementation Deatils.*** We implemented the proposed ANEL using the PyTorch framework and the PyG library on a node with A800 GPUs. For optimization, we chose the Adam optimizer with a learning rate set to 0.001, and the embedding dimensions were fixed at 200.

For hyperparameter tuning in the ANEL, we used grid search on the validation set to ensure optimal performance. The search range for k was 3, 6, 9, 12, 15, 18, 21 and the dropout parameter was searched within 0.2, 0.4, 0.6. For the four benchmark datasets employed in the experiment—ICEWS14, ICEWS05-15, ICEWS18, and GDELT—the optimal k values for REGCN-ANEL and TiRGN-ANEL were determined to be 6, 9, 6, 12, and 15, 15, 3, 9, respectively. Furthermore, a dropout rate of 0.4 was applied in each layer. With

Table 2: ICEWS14, ICEWS05-15, ICEWS18 and GDELT. Best results are in bold.

| Model | ICEWS14 | | | ICEWS05-15 | | | ICEWS18 | | | GDELT | | |
|---|---|---|---|---|---|---|---|---|---|---|---|---|
| | MRR | Hit@1 | Hit@10 | MRR | Hit@1 | Hit@10 | MRR | Hit@1 | Hit@10 | MRR | Hit@1 | Hit@10 |
| DistMult [44] | 25.31 | 17.83 | 42.20 | 18.39 | 11.16 | 30.32 | 16.59 | 10.01 | 31.69 | 15.64 | 9.37 | 29.33 |
| ConvE [45] | 31.23 | 21.20 | 50.37 | 30.40 | 20.21 | 49.96 | 24.16 | 15.47 | 44.32 | 17.03 | 10.21 | 33.17 |
| RotatE [46] | 27.41 | 18.53 | 47.46 | 19.47 | 10.32 | 38.91 | 15.33 | 6.90 | 32.82 | 5.43 | 1.87 | 13.64 |
| ComplEx [47] | 31.79 | 22.91 | 51.43 | 22.55 | 14.22 | 40.96 | 18.76 | 11.39 | 25.73 | 12.07 | 8.24 | 20.29 |
| TNTComplEx [17] | 30.02 | 23.33 | 49.11 | 27.52 | 19.50 | 42.85 | 21.21 | 13.26 | 36.89 | 19.51 | 12.38 | 33.39 |
| RE-NET [48] | 38.21 | 28.43 | 54.11 | 42.77 | 30.89 | 63.29 | 28.83 | 19.07 | 47.53 | 19.65 | 12.45 | 34.07 |
| CyGnet [49] | 37.32 | 27.14 | 57.49 | 39.95 | 28.91 | 60.96 | 26.35 | 16.23 | 44.71 | 19.72 | 11.83 | 33.21 |
| xERTE [50] | 40.69 | 32.65 | 57.26 | 46.57 | 37.76 | 63.87 | 29.28 | 21.01 | 45.56 | 18.75 | 11.92 | 32.45 |
| TITer [51] | 41.70 | 32.71 | 58.44 | 47.61 | 38.33 | 64.89 | 29.97 | 22.04 | 44.86 | 18.20 | 11.62 | 31.23 |
| REGCN [11] | 41.75 | 31.59 | 61.46 | 46.44 | 35.99 | 66.45 | 32.2 | 22.08 | 51.98 | 19.63 | 12.34 | 33.78 |
| TiRGN [43] | 43.67 | 33.09 | 62.21 | 49.36 | 38.73 | 69.72 | 33.26 | 22.87 | 53.64 | 21.66 | 13.61 | 37.68 |
| CENET [52] | 41.25 | 32.23 | 58.16 | 47.05 | 37.16 | 67.57 | 29.63 | 19.87 | 48.12 | 19.67 | 11.94 | 34.96 |
| REGCN-ANEL (Ours) | 43.28 | 32.93 | 63.11 | 49.22 | 38.4 | 69.89 | 33.57 | 23.15 | 53.31 | 20.38 | 12.88 | 35.02 |
| TiRGN-ANEL (Ours) | **44.74** | **34.4** | **64.66** | **50.59** | **39.69** | **71.27** | **34.04** | **23.53** | **54.71** | **22.06** | **13.85** | **38.33** |

Table 3: Performance comparison of REGCN and REGCN + ANEL (Ours) on Benchmark datasets.

| Dataset | Model | MRR | H@1 | H@3 | H@10 |
|---|---|---|---|---|---|
| ICEWS14 | REGCN | 41.75 | 31.59 | 46.43 | 61.46 |
| | REGCN-ANEL (Ours) | **43.28** | **32.93** | **48.03** | **63.11** |
| | △ improve | 3.67% | 4.24% | 3.45% | 2.69% |
| ICEWS18 | REGCN | 32.20 | 22.08 | 36.25 | 51.98 |
| | REGCN-ANEL (Ours) | **33.57** | **23.15** | **37.40** | **53.31** |
| | △ improve | 4.26% | 4.85% | 3.17% | 2.56% |
| ICEWS05-15 | REGCN | 46.44 | 35.99 | 52.08 | 66.45 |
| | REGCN-ANEL (Ours) | **49.22** | **38.4** | **55.15** | **69.89** |
| | △ improve | 5.99% | 6.7% | 5.9% | 5.18% |
| GDELT | REGCN | 19.63 | 12.34 | 20.91 | 33.78 |
| | REGCN-ANEL (Ours) | **20.38** | **12.88** | **21.77** | **35.02** |
| | △ improve | 3.82% | 4.38% | 4.12% | 3.67% |

Table 4: Performance comparison of TiRGN and TiRGN + ANEL (Ours) on Benchmark datasets.

| Dataset | Model | MRR | H@1 | H@3 | H@10 |
|---|---|---|---|---|---|
| ICEWS14 | TiRGN | 43.67 | 33.09 | 48.11 | 62.21 |
| | TiRGN-ANEL (Ours) | **44.74** | **34.4** | **49.82** | **64.66** |
| | △ improve | 2.5% | 3.96% | 3.55% | 3.94% |
| ICEWS18 | TiRGN | 33.26 | 22.87 | 37.58 | 53.64 |
| | TiRGN-ANEL (Ours) | **34.04** | **23.53** | **38.41** | **54.71** |
| | △ improve | 2.35% | 2.89% | 2.21% | 2.0% |
| ICEWS05-15 | TiRGN | 49.36 | 38.73 | 55.27 | 69.72 |
| | TiRGN-ANEL (Ours) | **50.59** | **39.69** | **56.67** | **71.27** |
| | △ improve | 2.49% | 2.48% | 2.53% | 2.22% |
| GDELT | TiRGN | 21.66 | 13.61 | 23.28 | 37.68 |
| | TiRGN-ANEL (Ours) | **22.06** | **13.85** | **23.76** | **38.33** |
| | △ improve | 1.85% | 1.76% | 2.06% | 1.73% |

these parameters, we present the experimental outcomes of our proposed ANEL model after integrating it with the baseline architecture. For the other comparison experiments, we used the default model parameters.

## 5.2 Main Results (RQ1)

Table 2 presents a comparison of various models' performance on entity prediction tasks within a benchmark dataset. To highlight the flexibility and effectiveness of our ANEL framework, we incorporated it into two leading TKGC models. We chose the TiRGN model, which achieved the highest entity prediction accuracy, and the REGCN model, which secured the second position. By integrating ANEL, we developed two hybrid models—TiRGN-ANEL and REGCN-ANEL—to evaluate their performance in entity prediction tasks.

As demonstrated in Tables 2, the TiRGN-ANEL and REGCN-ANEL models significantly surpass static models in entity prediction tasks, primarily because static models are incapable of capturing essential temporal dynamics in KGC. Moreover, both TiRGN-ANEL and REGCN-ANEL exceed their original base architectures. This advancement is largely attributable to the REGCN model's dependency on the most recent entity representations, which overlooks wider global dependencies that evolve over time. While the TiRGN model accounts for these global dependencies, its approach remains somewhat constrained. ANEL refines this by dynamically adjusting the integration of global dependencies based on the number of neighboring entities. In contrast, models like RE-NET and CyGNET underperform due to their inability to fully leverage temporal information within the same time step. Likewise, xERTE and TITer, which depend on path-searching strategies for entity prediction, struggle when critical paths are absent for certain entities.

Tables 3 and 4 illustrate the performance enhancements in MRR and H@{1,3,10} metrics for entity prediction following the integration of the proposed ANEL with the baseline models. The findings indicate that the REGCN-ANEL model increases the average entity prediction accuracy by 4.13% across four benchmark datasets, surpassing the base REGCN model, while TiRGN-ANEL achieves

a 2.54% improvement. The variance in accuracy gains arises from the fact that the TiRGN model leverages not only the original TKG but also historical interaction data between entities as input. Consequently, the TiRGN model has already incorporated latent information to some degree to enhance the original TKG, whereas the REGCN model relies solely on the TKG data as input.

## 5.3 Facts Sparsity Study (RQ1)

To evaluate the influence of data sparsity on the predictive performance of our TiRGN-ANEL and REGCN-ANEL models, we randomly removed between 10% and 90% of the facts from the ICEWS14 training set and assessed the models using the complete test set. Figure 3 provides a comparative analysis of MRR and Hits@10 in entity prediction tasks across four models: the ANEL-augmented TiRGN and REGCN, alongside the top-performing base models, TiRGN and REGCN. As the figure shows, the performance of all models declines with reduced training data. However, our TiRGN-ANEL and REGCN-ANEL models consistently outperform their base counterparts, exhibiting greater robustness to fact sparsity due to the integration of the ANEL mechanism.

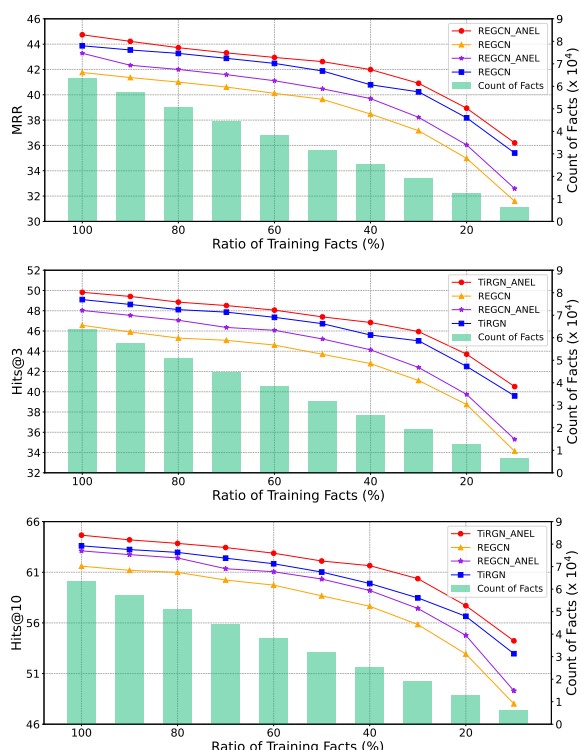

**Figure 3: The performance comparison of entity prediction between the ANEL-enhanced REGCN/TiRGN completion model and the base model on the sparsified ICEWS14 dataset.**

## 5.4 Sensitivity Analysis (RQ2)

In the proposed ANEL, the parameter $k$ determines the number of neighbors associated with the current entity in the latent graph, which is built upon filtered neighbors. The relationships between these neighbors in the latent graph are essential for updating the current entity's representation. To examine the effect of different $k$ values on the entity prediction MRR for the ANEL model and the base models REGCN/TiRGN, we conducted experiments with various $k$ values (3, 6, 9, 12, 15, 18, 21) to observe their impact on model performance (When k is 0, it refers to the prediction result of the baseline model). As depicted in Figures 4 and 5, the results demonstrate how varying $k$ influences the model's MRR metric.

The experimental results reveal that as the value of $k$ increases, the MRR of the ANEL fusion completion model initially rises and then declines. This is because, at lower $k$ values, a moderate number of potential neighbors provides valuable information, enhancing the representation of the current entity and improving model performance. However, when $k$ exceeds a certain threshold, the inclusion of too many potential neighbors introduces noise, which deteriorates the quality of the entity representation. This finding underscores that while the ANEL fusion completion model is adept at extracting rich information from potential neighbors to strengthen entity representation, careful selection of the number of neighbors is essential to prevent noise interference and sustain the model's effectiveness.

## 5.5 Ablation Study (RQ3)

The integration of the ANEL model with diverse base architectures surpasses the baseline models in entity prediction tasks across standard benchmark datasets. To evaluate the contribution of each component within ANEL, we performed an ablation study. In particular, we sequentially omitted the latent neighborhood mining component, the adaptive information module, the latent neighborhood aggregation module, and the attention mechanism from the REGCN-ANEL and TiRGN-ANEL models, assessing these modified versions on four datasets: ICEWS14, ICEWS05-15, ICEWS18, and GDELT.

As illustrated in Tables 5 and 6, the fully integrated ANEL model, when paired with the base models, delivers superior performance compared to the ANEL variants with individual components removed, underscoring the significance of each module. Notable observations include:

- **Latent Neighbourhood Mining Component:** We employ TiRGN and REGCN, two top-performing models for entity prediction in TKGC, as baseline models, incorporating the latent neighborhood mining (LNM) component introduced in this study. Across four benchmark datasets, all baseline models augmented with LNM substantially outperformed their counterparts lacking the latent information mining component (w/o LNM). This highlights that the LNM component offers valuable supplementary insights for TKGC tasks, enhancing the models' capability to capture essential temporal relationship dependencies.
- **Adaptive information Component:** The complete ANEL model also surpasses the variants without the Adaptive Information component (w/o AI), suggesting the necessity of a regulatory mechanism that gauges the need for latent relational data. This is especially crucial for sparse entities with fewer relational connections, as they gain from the supplementary information offered by this component, leading to more precise predictions.

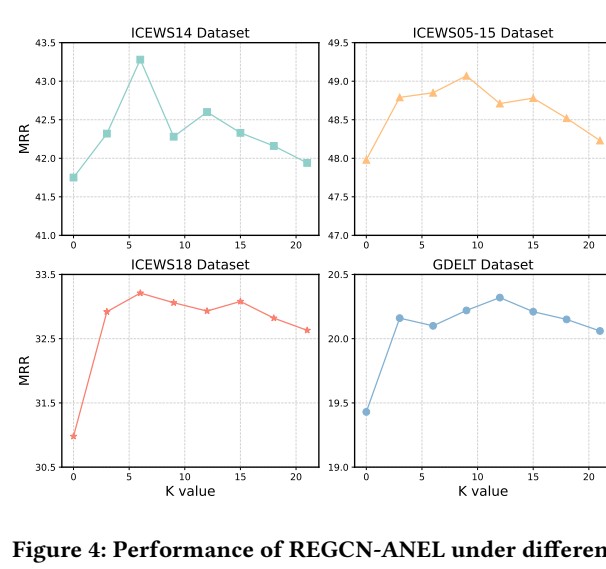

**Figure 4: Performance of REGCN-ANEL under different $k-$ value innce in the ICEWS14, ICEWS18, ICEWS05-15, and GDELT benchmark.**

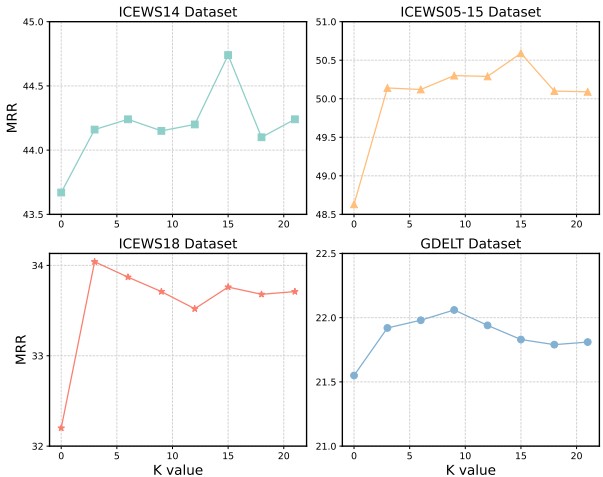

**Figure 5: Performance of TiRGN-ANEL under different $k-$ value innce in the ICEWS14, ICEWS18, ICEWS05-15, and GDELT benchmark.**

- **Latent Neighbourhood Aggregation Component:** The Latent Neighborhood Aggregation (LNA) component strengthens the model's capacity to represent sparse entities by extracting potential neighborhood information. To validate the LNA component's impact on entity prediction tasks, we excluded it from the model and compared the prediction performance across various datasets with the complete model. The results revealed a notable decline in accuracy when the LNA component was omitted (w/o LNA), confirming that LNA aids the model in capturing additional latent temporal relationships by enriching neighborhood information, thereby boosting overall prediction performance.
- **Attention Mechanism Component:** To assess the contribution of the attention mechanism, we excluded it and examined its impact on the performance of the base models integrated with ANEL (w/o AM). The results indicate that the full model outperforms the variant lacking the attention mechanism, confirming that the attention mechanism effectively identifies and distinguishes the significance of various latent entities. This differentiation enhances the base models' capacity to deliver more accurate entity predictions.

**Table 5: Performance of different variants of REGGN-ANEL on various datasets**

| Model | ICEWS14 | ICEWS05-15 | ICEWS18 | GDELT |
|---|---|---|---|---|
| w/o LNM | 41.75 | 46.44 | 32.2 | 19.63 |
| w/o AI | 42.24 | 48.48 | 32.67 | 20.21 |
| w/o LNA | 42.03 | 48.44 | 32.82 | 20.08 |
| w/o AM | 42.39 | 48.64 | 32.93 | 20.12 |
| REGGN-ANEL | 43.28 | 49.22 | 33.57 | 20.28 |

## 6 CONCLUSION

In this paper, we introduce ANEL to address the pervasive issue of sparsity in TKGC tasks. Our approach begins by generating an

**Table 6: Performance of different variants of TiRGN-ANEL on various datasets**

| Model | ICEWS14 | ICEWS05-15 | ICEWS18 | GDELT |
|---|---|---|---|---|
| w/o LNM | 43.67 | 49.39 | 33.26 | 21.66 |
| w/o AI | 42.52 | 50.14 | 33.77 | 21.83 |
| w/o LNA | 44.07 | 49.92 | 33.54 | 21.81 |
| w/o AM | 44.17 | 49.89 | 33.39 | 21.73 |
| TiRGN-ANEL | 44.71 | 50.59 | 34.04 | 22.06 |

initial embedding for each entity through a base model. We then incorporate a latent relation module that captures supplementary latent information for entities inadequately defined by their observed relationships. This is followed by an adaptive latent information component, which dynamically adjusts the level of latent information based on the observed data: entities with fewer direct relations are enriched with additional latent information, while those with denser relational data rely less on latent augmentation. The refined entity embeddings are then utilized for entity prediction tasks. Extensive experiments and analyses across four benchmark datasets demonstrate the efficacy and superiority of our proposed ANEL in TKGC.

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
