# OpenReview forum: "Tackling Sparse Facts for Temporal Knowledge Graph Completion"
_ACM.org/TheWebConf/2025/Conference — WWW 2025 Poster_

### Official Review · Reviewer_rVmN · 2024-11-28

**Novelty:** 5
**Technical Quality:** 4

**Review:**

This paper addresses fact sparsity in TKGC by proposing ANEL. Using four benchmark datasets and two metrics, along with comparisons to diverse baselines, is good, yet only integrating with two base models is limiting. The analyses of data sparsity, k-sensitivity, and ablation study are valuable, though performance improvement is sometimes modest. The paper is clear, and ANEL is novel, differing from prior methods and offering a new approach for TKGC via latent relation learning and adaptive adjustment, but could be enhanced with more base model integrations and further optimization.

Pros:
* Novel approach to address fact sparsity in TKGC.
* Comprehensive experimental evaluation with multiple datasets and baselines.
* The adaptive mechanism in ANEL is effective in handling entities with different degrees of sparsity.

Cons (detailed in the Questions section):
* The complexity of some concepts and equations might make it less accessible to a broader audience.

**Questions:**

1. In the ablation study, ANEL components contribute to performance. But what are the interactions between them? E.g., how does latent neighborhood mining affect the adaptive information component and vice versa?
2. The performance improvement is modest in some cases. What limits significant enhancement? What future research can overcome this, like exploring other latent info or better integrating ANEL with different base models?
3. The paper focuses on extrapolation in TKGs. What are the challenges and adaptations of ANEL for the interpolation task?

**Reviewer Confidence:**

2: The reviewer is willing to defend the evaluation, but it is likely that the reviewer did not understand parts of the paper

**Scope:**

4: The work is relevant to the Web and to the track, and is of broad interest to the community

---

### Official Review · Reviewer_KZ1o · 2024-11-28

**Novelty:** 5
**Technical Quality:** 5

**Review:**

This paper proposes the Adaptive Neighborhood Enhancement Layer (ANEL), a novel, model-agnostic module designed to address the pervasive issue of sparsity in Temporal Knowledge Graphs (TKGs). The quality of the work is high, as demonstrated by its rigorous experimentation across four benchmark datasets, comprehensive analyses (e.g., ablation and sensitivity studies), and robust empirical validation of its claims. It is highly original in its integration of latent neighborhood mining and adaptive information adjustment, providing a fresh approach to enhancing sparse entity representations. The work is significant, offering practical solutions for improving prediction accuracy in real-world tasks like event forecasting and recommendation systems.

Pros:

1.Innovative solution to the sparsity problem in TKGs.

2.Thorough experimentation and validation, showcasing improvements in metrics like MRR and Hits@10.

3.Broad applicability across various TKGC models.

4.Clear articulation of contributions and systematic evaluation.

Cons:

1.Scalability to larger, real-world datasets is not fully explored.

2.Certain technical sections, such as latent graph construction, rely on standard metrics like Pearson similarity, which might oversimplify complex relationships in TKGs.

Overall, this work makes a valuable contribution to the fields of temporal reasoning and knowledge representation.

**Questions:**

Q1.Are there specific challenges or constraints when integrating ANEL with non-GCN-based TKG models?

Q2.While ANEL improves representation by adding latent neighbors, the paper does not thoroughly explore scenarios where noisy or irrelevant neighbors might degrade performance.

Q3.How does the ANEL perform on larger, real-world datasets that exceed the benchmarks in size and complexity?

**Reviewer Confidence:**

3: The reviewer is confident but not certain that the evaluation is correct

**Scope:**

4: The work is relevant to the Web and to the track, and is of broad interest to the community

---

### Official Review · Reviewer_PC6m · 2024-11-29

**Novelty:** 2
**Technical Quality:** 5

**Review:**

The authors propose the method ANEL for temporal knowledge graph (TKG) completion that creates a latent neighbour graph. While code is provided, baselines are outperformed and the model serves as a plug-in for other models, there are issues with notation, novelty and others:
- Motivation: It is not clearly motivated (e.g., in the Introduction) why ANEL is specifically relevant for temporal KG completion. Fact sparsity occurs also in non-temporal KGs and neighboured node features are also available there.
- Notation: The notation used in problem statement and approach is not ideal.  For example, the prediction task is first mentioned in Section 3 but only defined (too late) in Section 4.1. $G_t$ is not defined in Section 3. The notation $C_{t_1}^s = (o_i \in G_{t_1})$ is unclear. In Equation 3, $\mathcal{G}^l$ is set to $0$ or $1$ and not to a set of edges.  $\mathcal{G}^l$  and $\mathcal{G}^{(l)}$  are used. In Figure 2, $g_n^l$ is not defined. At the end of Section 4, it is not clear what means “entities that have interacted”. Also, the problem statement is probably based on related works that should be cited.
- Novelty: Large parts of the methodology are, as denoted, taken from [39] and [41], leaving the latent graph construction as the only original contribution.
- Baselines: Several more recent baselines are missing such as L$^2$TKG (Zhang et al., 2023), LogCL (Chen et al., 2024) and RPC (Liang et al., 2023).
- Evaluation: Increase compared to the base models are rather small (e.g., from 43.67 to 44.74).
- Robustness: According to Section 5.3, ANEL shows greater robustness when more and more facts are removed. However, Figure 3b) shows that the margin between REGCN and REGCN-ANEL is actually decreasing when facts are removed.
- Code: The instructions in the readme stop at training. While I assume testing is included, this should be clarified. [Post-rebuttal] It is and documentation will be extended.
- Minor:
  - Figure 2 is never mentioned in the text.
  - Figure 2: “Eecoder”
  - Captions of Figure 4 and 5 have errors.

Post-Rebuttal comment: I acknolwedge the reply.

**Questions:**

1. Why is ANEL specifically suited for temporal KG completion (in contrast to non-temporal)?
2. Figure 3b) shows that the margin between REGCN and REGCN-ANEL is actually decreasing when facts are removed. Does this really confirm the robustness of ANEL?
3. Does the code include testing?

**Reviewer Confidence:**

3: The reviewer is confident but not certain that the evaluation is correct

**Scope:**

2: The connection to the Web is incidental, e.g., use of Web data or API

---

### Official Review · Reviewer_r3cp · 2024-12-03

**Novelty:** 5
**Technical Quality:** 5

**Review:**

In this paper, the authors focus on the link prediction task in temporal knowledge graph. They propose ANEL (Adaptive Neighborhood Enhancement Layer), a module agnostic to the underlying graph completion model that aims at improving the representation of sparse entities, i.e., entities that have a reduced number of neighbors.

The task of link prediction in temporal knowledge graph is well-studied and of importance. The problem of sparsity is also recognized in the community. I particularly appreciate the effort put by the authors to illustrate this problem (Fig 1) and propose a module that is agnostic to the underlying completion model, allowing to plug it on a wide variety of model. Overall, the paper provides a strong motivation and extensive experiments that showcase an increase in performance brought by the proposed module. I also appreciate the availability of the source code.

However, I have several concerns, in terms of presentation and scientific comprehension and discussion:
- The performance improvement appearing in Table 1 is more or less of 1 points in MRR. However, the increased complexity brought by including ANEL on top of a KGC model is never introduced (number of parameters, training time) and never discussed. At what costs is this improvement brought?
- Given the limited improvement, I think statistical significance tests could have strengthened the claim to "significantly surpass" SOTA approaches

- Some terms are used but never clearly defined, hindering the overall comprehension of the paper, e.g., "latent neighbor information", "structural intricacies and contextual semantics of latent entity relation".

- The definition of the link prediction task in Section 3 seems to differ from the examples in the introduction. Indeed, in the introduction temporal triples appears to be <h, r, o, t> with t a timestamp, whereas in section 3 the task seems to be to predict <h, r, ?, t> where t is a timestamp associated with a graph, meaning that the graph itself is dynamic [1]. This should clarified.

- The "latent graph construction" is not clear to me. First, it does not appear to be latent, since edges are actually selected in the actual graph: "set of potentially related entities by *filtering out* those that do not directly interact with entity s at the current timestamp". This means that you only select as o_i entities that directly interact with s at the current timestamp? And then select the top-k for later enrichment of the embedding of s? I interpreted "interact" as "a direct edge exist between s and o_i at timestamp t" but maybe I'm wrong. In any case, this should be clarified.

- Equation (4) and (5) are unclear to me. As the focus seems to be on o in (4), Equations (5) are strange:
-- w_sro = \frac{}{w \in \mathcal{N}_s}, shouldn't it be \mathcal{N}_o as we are computing the importance of inbound edges to o? (Equation 4)?
-- "while N_o denotes the set of entities that have interacted with entity s in the same graph", shouldn't it be entity o?

- Section 5.3: I was surprised that this ablation study is performed by removing triples from the train set. Couldn't have it been performed by showcasing a de-correlated MRR output by the model w.r.t. number of neighbors, augmenting Figure 1 with results outlining the de-correlation brought by ANEL?

Minor comments:
- the DACHA [22] - missing "model"
- the TeMP - missing "model"
- the REGCN - missing "model"
- Figure 2 is never referenced in text
- Baseline models TiRGN and REGCN augmented with ANEL are described in a paragraph "Evaluation Metrics". This is strange, this should be described in another paragraph.
- Caption of the top subfigure in Figure 3 appears wrong to the best of my knowledge as TiRGN should appear instead of REGCN for all lines
- Module names mentioned in 5.5 (latent neighborhood mining, adaptive information module, latent neighborhood aggregation, attention mechanism) should clearly appear and be described in Section 4 which is not the case for all of them.

[1] On a Generalized Framework for Time-Aware Knowledge Graphs. Krause et al.

**Questions:**

Could you comment on the following remarks?
- The performance improvement appearing in Table 1 is more or less of 1 points in MRR. However, the increased complexity brought by including ANEL on top of a KGC model is never introduced (number of parameters, training time) and never discussed. At what costs is this improvement brought?
- Given the limited improvement, I think statistical significance tests could have strengthened the claim to "significantly surpass" SOTA approaches

- The "latent graph construction" is not clear to me. First, it does not appear to be latent, since edges are actually selected in the actual graph: "set of potentially related entities by *filtering out* those that do not directly interact with entity s at the current timestamp". This means that you only select as o_i entities that directly interact with s at the current timestamp? And then select the top-k for later enrichment of the embedding of s? I interpreted "interact" as "a direct edge exist between s and o_i at timestamp t" but maybe I'm wrong?

- Equation (4) and (5) are unclear to me. As the focus seems to be on o in (4), Equations (5) are strange:
-- w_sro = \frac{}{w \in \mathcal{N}_s}, shouldn't it be \mathcal{N}_o as we are computing the importance of inbound edges to o? (Equation 4)?
-- "while N_o denotes the set of entities that have interacted with entity s in the same graph", shouldn't it be entity o?

**Reviewer Confidence:**

3: The reviewer is confident but not certain that the evaluation is correct

**Scope:**

3: The work is somewhat relevant to the Web and to the track, and is of narrow interest to a sub-community

---

### Official Review · Reviewer_PgUc · 2024-12-04

**Novelty:** 4
**Technical Quality:** 4

**Review:**

The paper introduces the Adaptive Neighborhood Enhancement Layer (ANEL) model for the enhancement of methods for Temporal Knowledge Graphs Completion (TKGC) in presence of sparse data.

The paper first revises the related approaches for TKGC and provide basic definitions for Temporal KGs and completion tasks. The authors then introduce the ANEL model, detailing the enhancement by the stages of latent graph construction, temporal neighbor sampling and neighbor aggregation. The method is evaluated by experiments verifying the effects of application of ANEL to known TKG datasets and different TKGC approaches.

In general, the problem of sparsity in KGs (and temporal KGs) representations is clearly an issue of interest to the KG community and the proposed ANEL model shows some benefits in the presented experimental results.

On the other hand, the contributions of the paper should be further detailed, also considering the broader audience of the conference: in the definition of the ANEL model, it is not clear what is the intuition behind the choices of the model and what is the effect of the temporal aspect in the managing of sparsity.

The technical quality of the paper is sufficient and the code is made available for replication.
The clarity of the paper should be revised to better understand the contributions from the point of view of the KG community.
The originality of the contributions (with respect to related approaches for sparsity as [39]) should be better discussed.
The work can be useful to the TKG community, but could be better related to the general WWW and KG topics.

PROS:
- The model shows advantages in the application to different temporal KGs and completion methods
- A number of TKGC approaches have been considered in the evaluation
- The ANEL model is shown to be applicable to different TKGC approaches

CONS:
- The role of the model in managing sparsity could be explained more in detail
- The writing of the paper is very technical and assumes that the reader is familiar with (embedding) methods for TKGC
- Results of the experimentation should be discussed in more detail, to better understand the effect of the model to the presented tasks

**Questions:**

- What is the main source of sparsity that ANEL is aims to solve?
- Can ANEL be applied to general (non-temporal) KGs? How much is the temporal aspect relevant to the model?

**Reviewer Confidence:**

2: The reviewer is willing to defend the evaluation, but it is likely that the reviewer did not understand parts of the paper

**Scope:**

3: The work is somewhat relevant to the Web and to the track, and is of narrow interest to a sub-community